Life history and ossification patterns in Miguashaia bureaui reveal the early evolution of osteogenesis in coelacanths

Mondéjar Fernández Jorge 1 2 jorge.mondejar-fernandez@mnhn.fr
Meunier François J. 3
Cloutier Richard 4
Clément Gaël 2
http://orcid.org/0000-0003-2974-9835 Laurin Michel 2
1 Division Paleontology and Historical Geology, Senckenberg Research Institute and Natural History Museum , Frankfurt am Main , Germany
2 Centre de Recherche en Paléontologie—Paris (CR2P), UMR 7207, MNHN, CNRS, SU, Département Origines et Évolution, Muséum National d’Histoire Naturelle , Paris , France
3 Laboratoire de Biologie des Organismes et des Écosystèmes Aquatiques (BOREA), UMR 8067, MNHN, CNRS, SU, Département Adaptations du Vivant, Muséum National d’Histoire Naturelle , Paris , France
4 Département de Biologie, Chimie et Géographie, Université du Québec à Rimouski , Rimouski , Canada
Abdala Virginia
Electronic publication date: 2022 Apr 6
Publication date: 2022
Volume: 10
Electronic Location ID: e13175
Received 2022 Jan 7; Accepted 2022 Mar 6
Copyright: © 2022 Mondéjar Fernández et al.
Copyright year: 2022
Copyright holder: Mondéjar Fernández et al.
License: This is an open access article distributed under the terms of the Creative Commons Attribution License, which permits unrestricted use, distribution, reproduction and adaptation in any medium and for any purpose provided that it is properly attributed. For attribution, the original author(s), title, publication source (PeerJ) and either DOI or URL of the article must be cited.
License URL: https://creativecommons.org/licenses/by/4.0/

Keywords: Bone, Cartilage, Evolution, Palaeobiology, Skeletochronology, Sarcopterygii, Devonian, Vertebrate

Funding: Louis Gentil-Jacques Bourcart prize of the French Academy of Sciences Natural Sciences and Engineering Research Council of Canada Jorge Mondéjar Fernández was supported by the Louis Gentil-Jacques Bourcart prize of the French Academy of Sciences. Richard Cloutier was funded by the Natural Sciences and Engineering Research Council of Canada (Discovery Grant). The funders had no role in study design, data collection and analysis, decision to publish, or preparation of the manuscript.

==============================
The study of development is critical for revealing the evolution of major vertebrate lineages. Coelacanths have one of the longest evolutionary histories among osteichthyans, but despite access to extant representatives, the onset of their weakly ossified endoskeleton is still poorly understood. Here we present the first palaeohistological and skeletochronological study of Miguashaia bureaui from the Upper Devonian of Canada, pivotal for exploring the palaeobiology and early evolution of osteogenesis in coelacanths. Cross sections of the caudal fin bones show that the cortex is made of layers of primary bone separated by lines of arrested growth, indicative of a cyclical growth. The medullary cavity displays remnants of calcified cartilage associated with bony trabeculae, characteristic of endochondral ossification. A skeletochronological analysis indicates that rapid growth during a short juvenile period was followed by slower growth in adulthood. Our new analysis highlights the life history and palaeoecology of Miguashaia bureaui and reveals that, despite differences in size and habitat, the poor endoskeletal ossification known in the extant Latimeria chalumnae can be traced back at least 375 million years ago.

Introduction

The retrogressive development of the endoskeleton is among the main trends identified in the long evolutionary history of coelacanths (Jarvik, 1980; Cloutier, 1991a, 1991b; Janvier, 1996; Forey, 1998), and includes not only the reduction in the number of bones but also the replacement of well-ossified elements by more cartilaginous ones in the skull and postcranium. The postcranial anatomy displays a relative morphological conservatism across coelacanth evolution, as shown by the general body shape, the appendicular skeleton, and the squamation (Schaeffer, 1948; Forey, 1998; Mondéjar-Fernández et al., 2021). This morphological stasis may be linked to locomotor constraints (Schaeffer, 1948; Cloutier, 1991a), while paedomorphosis has been cited as the primary process responsible for reductive trends (Lund & Lund, 1985; Forey, 1998).

The braincase, axial skeleton and fins of coelacanths are composed of endoskeletal elements (e.g., neurocranium, vertebrae, mesomeres and radials) covered by dermoskeletal products (e.g., dermocranium, scales, fin rays). In the extant Latimeria chalumnae, the endoskeleton shows a reduced ossification rate, especially in the neurocranium (Millot & Anthony, 1958; Forey, 1998; Dutel et al., 2019). Similarly, the paired (pectoral and pelvic) and median (anterior and posterior dorsal, anal, and caudal) fins are essentially made of cartilage with a limited contribution of endochondral ossification (Francillon et al., 1975; Mansuit et al., 2019, 2021). Other bony tissues have been histologically surveyed, such as the mineralized plates surrounding the lung in extinct (Brito et al., 2010) and extant taxa (Cupello et al., 2017a, 2017b; Meunier et al., 2021). However, contrary to the dermoskeleton (Smith, Hobdell & Miller, 1972; Castanet et al., 1975; Giraud et al., 1978; Meunier, 1980; Meinke, 1982; Meunier & Zylberberg, 1999; Meunier et al., 2008; Mondéjar-Fernández et al., 2021), few histological studies have been carried out on the coelacanth endoskeleton (Francillon et al., 1975), thus limiting our knowledge on its microstructure and evolution.

Miguashaia bureaui from the Upper Devonian Miguasha Lagerstätte in Québec (Canada) is one of the morphologically primitive coelacanths (Schultze, 1973; Cloutier, 1991b, 1996; Forey et al., 2000; Zhu et al., 2012) and thus key to the understanding of their early evolution and lifestyle. Studies using palaeohistology as a proxy for reconstructing the palaeobiology of Devonian sarcopterygians are rare, with a few exceptions mainly focused on taxa close to the emergence of tetrapods (Laurin et al., 2007; Sanchez, Tafforeau & Ahlberg, 2014; Kamska et al., 2018). The endoskeleton of Miguashaia bureaui has never been previously histologically surveyed, although the rich and diverse material from Miguasha constitutes an extraordinary source of information (Janvier & Arsenault, 2002; Janvier, Arsenault & Desbiens, 2004; Downs & Donoghue, 2009; Zylberberg, Meunier & Laurin, 2010; Meunier & Laurin, 2012; Zylberberg, Meunier & Laurin, 2015; Chevrinais, Sire & Cloutier, 2017). Bone microstructure is informative on numerous life history traits of vertebrates, from metabolism, to growth rate, and physiology (Francillon-Vieillot et al., 1990), and these can be inferred from the density, size, and shape of the cellular bone lacunae and the intensity of vascularization, among others (Amprino, 1947). Moreover, skeletochronology provides detailed information about growth, individual age, and sexual maturity based on natural and repetitive marks on the bone (Castanet, Meunier & de Ricqles, 1977; Castanet et al., 1993; Geffen et al., 2002; Meunier et al., 2002; Morales-Nin & Panfili, 2002).

Here we investigate the palaeohistology of the median fin bones (radials) of the Devonian coelacanth Miguashaia bureaui and compare it with the extant Latimeria chalumnae. Our main objectives are: 1. to describe bone microstructure with respect to life history traits in Miguashaia bureaui; 2. to evaluate the importance of endoskeletal ossification in the coelacanth postcranium and retrace its evolution; and 3. to elucidate the palaeoecology of one of the earliest coelacanths.

Materials and Methods

Miguashaia bureaui occurs in the 375 Myr-old (middle Frasnian, Upper Devonian) Escuminac Formation, Miguasha, Québec, Canada. Fossil material used for the histological investigation is housed at the Musée d’Histoire naturelle de Miguasha (MHNM) and consists of a block (MHNM 06-1238, Fig. 1) containing scales and disarticulated elements (Mondéjar-Fernández et al., 2021) associated with radial bones, most likely from the caudal fin of a single individual given the partial articulation and similar relative size of the osseous remains. Based on scale diameter (ca. 25 mm), our studied specimen (MHNM 06-1238) has an estimated total length (TL) of approximately 400 mm.

Figure 1 Anatomy and fossil material of Miguashaia bureaui.

(A) Reconstruction of Miguashaia bureaui (Cloutier, 1996). (B) Fossil material of Miguashaia bureaui used for the histological investigation (MHNM 06-1238). The block contains numerous articulated scales (sc) associated with disarticulated axial and appendicular bones of a single individual. Caudal fin radial bones (white arrows) were extracted from specimen MHNM 06-1238 before and during its preparation. Abbreviations: a.f, anal fin; c.f, caudal fin; d.f1-2, dorsal fins; hs, haemal spine; lp, lepidotrichia; mc, medullary cavity; ns, neural spine; pc.f, pectoral fin; pv.f, pelvic fin; ra, radials; sc, scale.

The fossil material was embedded in a polyester resin and sectioned after 48 h of drying in an incubator at 40 °C. The selected sections were cut with a diamond saw, ground on a diamond plate, polished with a suspension of alumina powder to a thickness of 60–80 μm, glued on a glass slide with 20/20 araldite, and observed under transmitted natural light with an Olympus BX51 microscope. Pictures were taken with a digital camera Olympus Camedia C-5060.

Comparative data on Latimeria chalumnae come from microradiographies (X-ray beam power: 10 to 15 KV at 7 to 8 mA; distance between the X-ray source and the specimens: 10 cm) of MNHN-ZA-AC-2012-26 (CCC79, adult female, 78 kg, 163 cm TL) (Francillon et al., 1975), housed in the Collection of Comparative Anatomy of the Muséum national d’Histoire naturelle (MNHN) of Paris, France.

All the fossil material of Miguashaia bureaui is housed at the Musée d’Histoire naturelle de Miguasha (MHNM) (32 specimens; updated count from Cloutier, Proust & Tessier (2011)), parc national de Miguasha, Québec (Canada) with the exception of the three specimens (including the holotype ULQ Esc 120) at Université Laval (Québec City, Québec,Canada), six specimens at The Natural History Museum (BMNH) (London, UK), and one specimen at the American Museum of Natural History (New York, USA).

Results

Morphology

The radials are cylindrical bones (between 1–2 cm in length and 2–3 mm in width) composed of a slightly narrowing median shaft (diaphysis) and two small, somewhat enlarged dorsal and ventral extremities (epiphyses) (ra, Fig. 1) (Cloutier, 1996). The bones are roughly circular in cross section and may present a lateral expansion (Figs. 2, 3). The epiphyses articulate with the ossified dermal fin rays (lp, Fig. 1A) and are open-ended in fossil specimens (Fig. 1B) (Cloutier, 1996), suggesting the occurrence of distal cartilaginous pads in the living animal.

Figure 2 Median fin bone palaeohistology of Miguashaia bureaui (MHNM 06-1238).

(A and B). Transversal cross sections of two radials. Black arrowheads indicate the lateral expansions of the bones. White arrowheads indicate LAGs (lines of arrested growth). Insets are detailed in Fig. 3.

Figure 3 Median fin bone palaeohistology of Miguashaia bureaui (MHNM 06-1238).

(A) Large zone with radial vascular canals (vc) and large osteocyte lacunae. (B) Axial vascular canals (vc) (i.e., primary osteons). (C) Detail of a wide (left) and several narrow (right) zones. (D) Inset of a zone with axial vascular canals (vc) and large osteocyte lacunae. (E) Detail of a wide zone (left) with axial vascular canals (vc) and large osteocyte lacunae and one of the first avascular narrow zones (right) with small osteocyte lacunae. (F) Detail of several tightening narrow zones. White arrow heads indicate the presumed course of the LAGs represented in Fig. 2. Black arrows with white heads indicate osteocyte lacunae. Abbreviations: vc, vascular canal.

Histology

Diaphyseal bone histology

Transversal (Figs. 2–4) and longitudinal (Figs. 5A, 5C, 5E–5H) sections reveal that the radials are composed of a relatively thick cortex surrounding a central medullary cavity (or medulla) that measures from 50 μm to 1 mm in diameter, depending on the location of the section across the diaphysis (Figs. 4A, 4D). The osseous tissue consists of primary periostic bone with superimposed layers of parallel-fibered or pseudo-lamellar bone. The enclosed star-shaped cell lacunae (i.e., osteocyte lacunae) (30–40 μm in length and 2, 5–5 μm in width) send ramified cytoplasmic processes (i.e., canaliculi) in a heterogenous extracellular matrix (Figs. 3, 4C, 4E, 4F, 5G, 5H). Their orientation is variable, from roughly radial close to the medulla (Figs. 3A, 3B, 3D, 3E, 4C) to parallel to the external surface approaching the periphery (Figs. 3C, 3F, 4F–4E). The bony layers are crossed by radial (Fig. 3A) or axial (Figs. 3B, 3D, 3E) vascular canals, the majority of which are narrow simple primary canals, but some of the axial ones are surrounded by concentric layers of bone lamellae (i.e., primary osteons) (Figs. 3B, 3E, 4D). The cortex shows substantial cyclicity (Figs. 2–4) evidenced by concentric lines of arrested growth (i.e., LAGs) separating the bone layers (i.e., zones). In the areas close to the medulla, the zones are thicker, display larger vascular canals, larger osteocyte lacunae, and are crossed by Sharpey’s fibres (Fig. 5H) whereas the zones close to the periphery are thinner, the number of vascular canals decreases drastically, and the osteocyte lacunae are fewer and smaller (Figs. 3C, 3E, 3F, 4B, 4E, 4F). Growth was highly asymmetrical, as shown by the eccentric location of the medullary cavity and the variable thickness of the zones (Fig. 2). The regular profile of the inner walls of the medulla indicates that little resorption of the cortex occurred, if any.

Figure 4 Median fin bone palaeohistology of Miguashaia bureaui (MHNM 06-1238).

(A, D) Transversal cross section of two radials and detailed insets (B and C, E and F, respectively). Note the difference in width of the medullary cavity (mc) between the mid-diaphysis (A) and near the epiphyses (D). (B) Peripheral layers of compact bone from the specimen in (A). Note the lack of vascularization and the reduction in the number of osteocyte lacunae. (C) Deeper layers of compact primary bone. Note the larger size of the osteocyte lacunae in comparison with more peripheral layers. (E and F) Peripheral layers illustrating a drastic reduction in the number of osteocyte lacunae and highlighting the putative location of the periosteal membrane or periosteum (dark brown). Black arrows with white heads indicate osteocyte lacunae. Abbreviations: mc, medullary cavity.

Figure 5 Comparative histology of extinct (Miguashaia bureaui) and extant (Latimeria chalumnae) coelacanths.

(A–H) Thin sections of Miguashaia bureaui (MHNM 06-1238) in transmitted light. Longitudinal (A, C) and transversal (B) cross section of three radials. (D–F) Remnants of calcified cartilage with mineralized cartilaginous spheritic structures (black arrowheads) and bony trabeculae (black arrows) around a bony cortex (white asterisks) with vascular canals (vc) evidencing localized remodeling processes of the primary bone (white arrowhead). (G) Detail of osteocyte lacunae with long cytoplasmic prolongations (canaliculi) (black arrows with white heads). (H) Detail of Sharpey’s fibres (white arrows with black heads). (I–K) Microradiographs of Latimeria chalumnae (MNHN-ZA-AC-2012-26) (Francillon et al., 1975). (I) Meckelian bone (white asterisks) containing fine bony trabeculae (white arrows) resulting from the erosion of the Meckelian cartilage. (J) Cortical region of the angular bone showing numerous vascular canals (vc), some of them with cementing lines (white arrowhead) and Sharpey’s fibers (black arrows with white heads). (K) Scapulocoracoid showing the resorption of the cartilage by canals (black asterisks) whose walls are covered by thin bony layers (in white).

Mineralized cartilage and endochondral ossification

The medulla was probably filled up, at least partially, by unmineralized cartilaginous tissue in the living animal, but most of it has not been fossilized. However, patches of mineralized cartilage can still be seen (Figs. 5A–5F). The mineralized structures are spheritic, with the crystals and the organic matrix showing a radial arrangement, and display thin increment lines (i.e., Liesegang rings), which separate the successive layers formed during calcification. Spherical chondrocyte lacunae reveal the location of the chondrocytes (Fig. 5F). The mineralized cartilage shows marks of erosion and has been replaced by thin bony trabeculae (Figs. 5D–5F), a phenomenon characteristic of endochondral ossification (Francillon-Vieillot et al., 1990). The occurrence of these patches of calcified cartilage also confirms the lack of medullary erosion of the cortex.

Skeletochronological analysis

The serial organization of thick vascularized bony layers followed by thin avascular layers is indicative of growth processes, with the wide zones representing areas of fast growth typical of the juvenile period, and the narrow zones corresponding to the slower growth of adulthood (Castanet, Meunier & de Ricqles, 1977; Castanet et al., 1993; Geffen et al., 2002; Meunier et al., 2002; Morales-Nin & Panfili, 2002; Schucht, Klein & Lambertz, 2021) (Fig. 2). Lines of arrested growth (LAGs) are structures expressing a pause of the osteogenesis during annual estivation and/or hibernation in poikilothermic vertebrates (Castanet, Meunier & de Ricqles, 1977; Castanet, 1985; de Ricqlès et al., 1991). One growth mark (i.e., one LAG and the immediately neighbouring layer), represents a seasonal and biological cycle. All the studied bones express simple LAGs, suggesting a single period of torpor with a likely annual frequency. Four wide zones followed by three thinner zones can be identified (Fig. 2). The absence of bone resorption accurately preserves the integrity of the growth marks and allows to confidently infer an age of seven years to the animal at the time of its death. A decrease in the growth-mark width towards the periphery indicates a possible acquisition of sexual maturity in its fifth year of life (Fig. 6A, Table 1).

Figure 6 Skeletochronology and size range of Miguashaia bureaui.

(A) Growth curves showing bone deposition rate in the cortex of two radials (a, b correspond to the bones illustrated in Figs. 2A, 2B and measured in Table 1) (MHNM 06-1238). The arrow points to the presumed acquisition of sexual maturity. (B) Histogram of the estimated total length of all the Miguashaia bureaui specimens and their frequency of occurrence in the Escuminac Formation (see Table 2). Profiles correspond to the juvenile morphotype (based on ULQ120; 197 mm TL) in grey and the adult morphotype (based on BMNH P.58691, BMNH P.62794, MHNM 06-41, MHNM 06-494; ca. 375 mm TL) in black (modified after Cloutier, 1996).

Table 1 Bone deposition in Miguashaia bureaui (MHNM 06-1238).

	0	1	2	3	4	5	6	7	
1 (Fig. 2A)	0	459	903	1,418	1,856	2,064	2,124	2,265	
2 (Fig. 2B)	0	162	366	582	913	1,033	1,150	1,247	
Note:

Measurements of cumulative radial bone deposition (μm) between the medullary margin and the various LAGs numbered from the inner part of the cortex to the periphery (i.e., from the first year of development (0) to the last year of life (7)).

Table 2 Complete specimens of Miguashaia bureaui.

Specimen no	MHNM 06-494	MHNM 06-1809A	MHNM 06-1236B	MHNM 06-41A	MHNM 06-1633	BMNH P. 58692 AB	BMNH P. 58693	BMNH P. 62794	ULQ Esc 120	BMNH P. 58691	MHNM 06-1318	MHNM 06-1810	BMNH P. 58694	MHNM 06-2414	
Total length (mm)	406.8	430.0	422.7	300.1	69.93	444.9	333.9	466.7	76.0	467.5	187.0	286.7	481.5	155.1	
Note:

Measurements of estimated total length (TL) of all the complete body specimens of Miguashaia bureaui from the Escuminac Formation.

Discussion

Our data confirm that in Miguashaia bureaui the increment in thickness of the radials resulted from the activity of an external osteogenic membrane (i.e., periosteum), which centrifugally deposited layers of primary bone across the diaphysis. These layers, separated by cementing LAGs, indicate a cyclical growth (Fig. 2). The distal elongation of the radials was likely due to the activity of chondroblasts as suggested by the combined occurrence of calcified cartilage and endochondral bony trabeculae in the medulla and close to the epiphyses (Figs. 5A–5F) (Meunier & Laurin, 2012). The calcification of cartilage took place in the form of hypertrophic cartilage (Haines, 1942) of spheritic type as observed in extant actinopterygians (e.g., the carp Cyprinus) (Zylberberg & Meunier, 2008). Remnants of calcified cartilage have also been found in other Devonian sarcopterygians like Eusthenopteron foordi (Laurin et al., 2007; Meunier & Laurin, 2012; Sanchez, Tafforeau & Ahlberg, 2014) and juveniles of Acanthostega gunnari (Sanchez et al., 2016). The retention of calcified cartilage in adult individuals of Miguashaia bureaui suggests a relatively modest endochondral ossification and slow epiphyseal growth. A poor endochondral ossification also occurs in the mainly cartilaginous paired fins of Latimeria chalumnae (Francillon et al., 1975; Mansuit et al., 2019, 2021), where the thin bony trabeculae show evidence of spheritic mineralized cartilage as well (Figs. 5I–5K). Moreover, in both Miguashaia bureaui and Latimeria chalumnae, bone remodelling is also limited to the inner walls of a few vascular canals and some endosteal bone deposition (Fig. 5).

The presence of fast and slow-growing osseous layers is characteristic of ectothermic vertebrates (de Ricqlès et al., 1991; Castanet, Francillon-Vieillot & de Ricqlès, 2003; Legendre et al., 2016; Legendre & Davesne, 2020). Their distribution in Miguashaia bureaui suggests that growth may have been indefinite but submitted to a pronounced cyclicity (such as seasonality), hinting at a poikilothermic metabolism. Moreover, a substantial reduction in thickness of the zones towards the periphery is generally explained by the acquisition of sexual maturity (Castanet et al., 1993; Meunier et al., 2002), which, despite that the reduction in deposition rate is not very drastic, points to the occurrence of the first reproduction in Miguashaia bureaui in the course of its fifth year (Fig. 6A). A short pre-reproductive period matches the range of some extant predatory osteichthyans of similar size to Miguashaia bureaui like bichirs (e.g., Polypterus senegalus) (Daget, Bauchot & Arnoult, 1965), sturgeons (e.g., Acipenser brevirostrum) (Kynard, 1997), gars (e.g., Lepisosteus osseus) (Smylie, Shervette & McDonough, 2016), dipnoans (e.g., Lepidosiren paradoxa) (Flower, 1935), as well as extant Batrachia (e.g., urodeles and anurans) (Castanet, Francillon-Vieillot & de Ricqlès, 2003; Amat & Meiri, 2018). A rapid growth and acquisition of maturity in Miguashaia bureaui are likely associated with the production of numerous offspring via external fertilization (Bone & Moore, 2008) and may also have enabled to quickly outgrow the larval and juvenile stages, more vulnerable to predation. However, it contrasts with the extremely late maturity revealed in Latimeria chalumnae (Mahé, Ernande & Herbin, 2021), which has been suggested to be related to internal fertilization and the production of a fully-developed offspring by viviparity after a long gestation period. The life history differences between Miguashaia bureaui and Latimeria chalumnae may reflect the more than 375 million years of evolution that separates both taxa but may also be due to their different sizes and habitats, with Miguashaia being a relatively small predator (Table 2, Fig. 6B) dwelling in moderately deep to shallow waters, while Latimeria is characterized by its large body size and is adapted to life in the deep waters of the mesopelagic zone where resources are less abundant (Forey, 1998; Cupello et al., 2017a; Mahé, Ernande & Herbin, 2021).

The differences in bone-growth rate identified in Miguashaia bureaui may also reveal differences in overall living conditions and changes in habitat across ontogeny. The Escuminac Formation represents a tropical, brackish, estuarine paleoenvironment (Schultze & Cloutier, 1996; Cloutier, Proust & Tessier, 2011) and most specimens come from laminated facies associated with tidal deposits. Miguashaia bureaui was suggested to be a predator within the Escuminac fauna (Chevrinais, Jacquet & Cloutier, 2017) but only a single specimen (MHNM 06-1237) preserves digestive content composed of a few of the spinicaudatan (conchostracan) Asmusia membranacea while six other specimens show solely amorphous organic matter. Miguashaia bureaui is only a minor component of the fossil assemblage representing 0.19% of the total abundance of vertebrate specimens (42 individuals out of 21,446 Escuminac vertebrate specimens). Of the 42 specimens, two could be considered juveniles measuring between 70 and 80 mm in length (Schultze, 1973; Cloutier, 1996), while rare specimens reached approximately 180 mm, and most other specimens are ca. 400–465 mm in total length (Table 2; Fig. 6B). One of the two juvenile specimens (MHNM 06-1633) was found in a nursery (or effective juvenile habitat) in association with hundreds of embryonic, larval, and juvenile individuals of the “placoderm” Bothriolepis canadensis, and the sarcopterygians Scaumenacia curta, Fleurantia denticulata and Eusthenopteron foordi. Although MHNM 06-1633 is relatively small for the species, it is one of the largest individuals found in the nursery. Juvenile specimens have not been found in association with larger adults. The Escuminac nursery may thus have been used by coelacanths, but most likely Miguashaia bureaui was not a permanent resident of the palaeoestuary, which suggests that the larger adult individuals inhabited and probably reproduced in more open and deeper marine waters, and only sporadically visited the Escuminac palaeoestuary.

The radials, neural and haemal spines of Miguashaia bureaui are among the sole endoskeletal postcranial elements retrieved so far, despite the good preservation of ontogenetic series from Miguasha (Cloutier, 1996, 2010), suggesting a truly low perichondral and endochondral ossification rate of the vertebral column and associated elements (Fig. 1A). A poorly ossified postcranium is common among early osteichthyans (Janvier, 1996), including coelacanths (Forey, 1998) such as Diplocercides kayseri from the Devonian of Germany, in which the radials appear to be only perichondrally ossified (Stensiö, 1937). The neural arches and spines of Laugia groenlandica from the Triassic of Greenland (Stensiö, 1932) also display a wide medullary cavity, possibly filled by cartilage, but the endoskeleton of the paired fins is surprisingly well ossified, as in Shoshonia arctopteryx from the Devonian of the USA (Friedman, Coates & Anderson, 2007). Many other extinct coelacanths are known from endoskeletal postcranial material (e.g., Forey, 1998; Long, 1999) but, to our knowledge, the ossification rate of the endoskeleton has not been palaeohistologically evaluated in these taxa.

Conclusions

Our data on the caudal fin radial bones of Miguashaia bureaui provide the first evidence of the potential of both perichondral and endochondral ossification in the postcranium of early coelacanths, with perichondral ossification being the main source of bone deposition and endochondral ossification having a minor contribution (Figs. 2, 5D–5F). This condition, also present in Latimeria chalumnae (Francillon et al., 1975; Meunier, Cupello & Clément, 2019) (Figs. 5I–5K), likely represents a retention of the primitive state for osteichthyans (Janvier, 1996), as suggested by the poorly ossified endoskeleton of close relatives of coelacanths, like onychodonts (Andrews et al., 2005; Mondéjar-Fernández, 2020). Miguashaia bureaui illustrates that, despite a different life history from that of Latimeria chalumnae, a postcranial osteogenetic pattern with reduced endochondral ossification and secondary bone remodelling has been maintained for the last 375 million years, showcasing a remarkable case of histological stasis in coelacanth evolution.

We warmly thank Severin Morel (MNHN) for the thin-sections, Philippe Loubry (MNHN) for the photographs, and Marc Herbin (MNHN) for granting access to the Latimeria collections under his care. Johanne Kerr (parc national de Miguasha) prepared the loan of specimens and Nathaniel Bertrand Maltais and Laurianne Richard (UQAR) helped with the photographs.

Additional Information and Declarations

Competing Interests

Author Contributions

Data Availability

The authors declare that they have no competing interests.

Jorge Mondéjar Fernández conceived and designed the experiments, performed the experiments, analyzed the data, prepared figures and/or tables, authored or reviewed drafts of the paper, and approved the final draft.

François J. Meunier conceived and designed the experiments, performed the experiments, analyzed the data, prepared figures and/or tables, authored or reviewed drafts of the paper, and approved the final draft.

Richard Cloutier conceived and designed the experiments, performed the experiments, analyzed the data, prepared figures and/or tables, authored or reviewed drafts of the paper, and approved the final draft.

Gaël Clément conceived and designed the experiments, authored or reviewed drafts of the paper, and approved the final draft.

Michel Laurin analyzed the data, authored or reviewed drafts of the paper, and approved the final draft.

The following information was supplied regarding data availability:

The raw data is available in Tables 1, 2 (measurements), Fig. 6 (graphs obtained from these measurements), Fig. 1 (a modified drawing with original information and fossil material), and Figs. 2–5 (histological photographs).

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
