# Peer review of "Life history and ossification patterns in Miguashaia bureaui reveal the early evolution of osteogenesis in coelacanths"

_PeerJ, doi:10.7717/peerj.13175_

## Round 0.1 · original submission · Minor Revisions

I received three reviews of your paper, all of them positive. As you will see, the suggestions are minor. However, many questions have been posited that need to be taken into account.

I look forward to your revised manuscript.

·

Basic reporting

This manuscript on the histology and its implications for the life history of one of the oldest known coelacanths is scientifically sound, well organized and well written. The list of references is complete, as far as I can tell, and the raw data is available. In fact, I suggest that the photograph of the complete specimen, now in the supplementary data, be included in the main text (and possibly also in the two small tables).

Experimental design

The methods for histological studies and data acquisition are well described and sufficient.

Validity of the findings

There are few histological studies on coelacanth taxa so far, except some on Latimeria. This study is important because it is one of the first on an extinct taxon. Moreover, it deals with one of the basal most genera of the lineage and brings important information on the life history of this fish (7 years at death and 5 years at sexual maturity). Additionally, he points out important characteristics of coelacanths in general, in particular their low rate of endoskeletal ossification which has remained constant for almost 400 million years (although other taxa have yet to be tested).

Reviewer 2 ·

Basic reporting

The manuscript by Mondéjar-Fernández et al. presents novel data on the ossification patterns in Miguashaia post-cranial structures (radials), which are then used to draw conclusions on the evolution of osteogenesis in coelacanths. In general, manuscript is clear, data are well presented, and conclusions are valid (to the best of my knowledge). English grammar is correct throughout the manuscript. Background knowledge is clearly presented in the introduction and well supported by citations to the literature. Objectives are also clearly defined and relevant. Materials and methods section, however, should be described with more details (see specific comments below). Results and Discussion are fine (see below for small adjustments).
Supplementary figure 1: Legend is missing. I would include this figure in the manuscript to clearly illustrate the starting material
Line 92: “from the caudal fin of a single individual” How can this be verified
Line 93: Give more details on how the material were sectioned and dried.
Line 94: Give more details on how the sections were ground and polished.
Line 97: Give more details on how the microradiographies were acquired.
Line 98: define TL.
Line 106: How were determine the length and width of the radials? From Figure S1?
Line 108: Figure 1A, not 1a
Line 109: lateral expansion should be indicated in figure 1
Line 110: S how evidence for “open-ended” radials.
Lines 169; 171; 181; 183; 191: Figures should be described in the results, not in the discussion.
Line 197: what is the ground for the association with the production of numerous offspring via external fertilization”?
Line 221: Scale measurement should be presented in the results.
Figure 1: Lettering in panels B and C is poorly visible
Figure 2: As in figure 1, lettering should be improved to be clearly visible. Same comment for the scale bars. Indicate osteocyte lacunae in the images.
Figure 5: define b and c in panel A

Experimental design

see comments above

Validity of the findings

see comments above

Additional comments

see comments above

·

Basic reporting

This manuscript provides informative paleohistological data of the caudal fin radials of a Late Devonian coelacanth, and reveals the osteogenesis in postcranial endoskeleton in early coelacanths.

Line 24: “weakly ossified skeleton”, here the authors should mean ‘endoskeleton’, even ‘postcranial endoskeleton’ The dermal skeleton and skull endoskeleton of coelacanths (at least for early coelacanths) is well ossified.
Line 25: Add ‘Upper’ in front of ‘Devonian’.
Line 36: ‘bones’ replaced by ‘radials’.
Line 33: poor rate?
Line 34: This sentence is somewhat misleading for the public (earliest coelacanths, 375 million years ago). Rewrite it.
Lines 41-44: The authors should be careful of the distinction between endoskeleton and dermal skeleton, as well as the distinction between endoskeleton of skull and postcranium. The mixed usage of these terms in the text will result in some logical inconsistency throughout the text. Did all categories of the skeleton undergo the retrogressive development in coelacanths? This should be clarified.
Line 52: important regression of the ossification rate?
Line 82: to evaluate the importance and evolution……? Rewrite it.

Figures: Median fin bones palaeohistology > Median fin bone palaeohistology?
References should be carefully edited.

Experimental design

No comment.

Validity of the findings

The authors should be careful of the caudal fin radials representing the endoskeleton of all categories (neurocranium, vertebrae, neural and haemal spines, endoskeletal girdles, etc.). Is it possible that different osteogenesis occurs in different category? Is it likely that ‘the postcranial osteogenetic pattern with reduced endochondral ossification and secondary bone remodelling’ is a plesiomorphy of coelacanths? I would suggest the authors impose more restrictions on their conclusions.

---

## Round 0.2 · Minor Revisions

Thank you for having the suggestions of our reviewers into consideration. I have one more point that requires your attention. Please note in the lines 220-221 that urodeles are not batrachians. Thus I suggest to rephrase that sentence as follows: as well as batrachians (e.g. anurans).

Reviewer 2 ·

Basic reporting

n.a.

Experimental design

n.a.

Validity of the findings

n.a.

Additional comments

Authors have satisfactorily addressed my comments and I consider that their manuscript can be published.

·

Basic reporting

The authors well addressed my previous concerns or comments.

Experimental design

no comment

Validity of the findings

no comment

Additional comments

no comment

---

## Round 0.3 · accepted · Accept

Thank you for considering my suggestion. I would be more comfortable using Batrachia instead of batrachians to avoid confusion with the colloquial term that is usually referred to as frogs. If this is OK for you, please use the formal name in capitals.